# COIN++: Neural Compression Across Modalities

**Emilien Dupont\***                                    *dupont@stats.ox.ac.uk*
**Hrushikesh Loya\***                                    *loya@stats.ox.ac.uk*
**Milad Alizadeh**                                 *milad.alizadeh@cs.ox.ac.uk*
**Adam Goliński**                                  *adamg@robots.ox.ac.uk*
**Yee Whye Teh**                                     *y.w.teh@stats.ox.ac.uk*
**Arnaud Doucet**                                    *doucet@stats.ox.ac.uk*
*University of Oxford*

**Reviewed on OpenReview:** *https://openreview.net/forum?id=NXBOrEM2Tq*

## Abstract

Neural compression algorithms are typically based on autoencoders that require specialized encoder and decoder architectures for different data modalities. In this paper, we propose COIN++, a neural compression framework that seamlessly handles a wide range of data modalities. Our approach is based on converting data to implicit neural representations, i.e. neural functions that map coordinates (such as pixel locations) to features (such as RGB values). Then, instead of storing the weights of the implicit neural representation directly, we store modulations applied to a meta-learned base network as a compressed code for the data. We further quantize and entropy code these modulations, leading to large compression gains while reducing encoding time by two orders of magnitude compared to baselines. We empirically demonstrate the feasibility of our method by compressing various data modalities, from images and audio to medical and climate data.

## 1 Introduction

It is estimated that several exabytes of data are created everyday (Domo, 2018). This data is comprised of a wide variety of data modalities, each of which could benefit from compression. However, the vast majority of work in neural compression has focused only on image and video data (Ma et al., 2019). In this paper, we introduce a new approach for neural compression, called COIN++, which is applicable to a wide range of data modalities, from images and audio to medical and climate data (see Figure 1).

Most neural compression algorithms are based on autoencoders (Theis et al., 2017; Ballé et al., 2018; Minnen et al., 2018; Lee et al., 2019). An encoder maps an image to a latent representation which is quantized and entropy coded into a bitstream. The bitstream is then transmitted to a decoder that reconstructs the image. The parameters of the encoder and decoder are trained to jointly minimize reconstruction error, or *distortion*, and the length of the compressed code, or *rate*. To achieve good performance, these algorithms heavily rely on encoder and decoder architectures that are specialized to images (Cheng et al., 2020b; Xie et al., 2021; Zou et al., 2022; Wang et al., 2022). Applying these models to new data modalities then requires designing new encoders and decoders which is usually challenging.

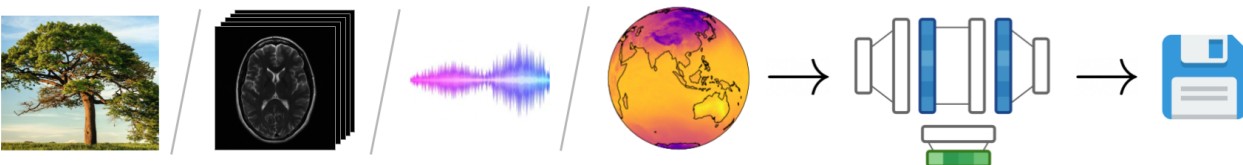

Figure 1: COIN++ converts a wide range of data modalities to neural networks via optimization and then stores the parameters of these neural networks as compressed codes for the data. Different data modalities can be compressed by simply changing the input and output dimensions of the neural networks.

Recently, a new framework for neural compression, called COIN (COmpression with Implicit Neural representations), was proposed which bypasses the need for specialized encoders and decoders (Dupont et al., 2021a). Instead of compressing images directly, COIN fits a neural network mapping pixel locations to RGB values to an image and stores the quantized weights of this network as a compressed code for the image. While Dupont et al. (2021a) only apply COIN to images, it holds promise for storing other data modalities. Indeed, neural networks mapping coordinates (such as pixel locations) to features (such as RGB values), typically called *implicit neural representations* (INR), have been used to represent signed distance functions (Park et al., 2019), voxel grids (Mescheder et al., 2019), 3D scenes (Sitzmann et al., 2019; Mildenhall et al., 2020), temperature fields (Dupont et al., 2021b), videos (Li et al., 2021b), audio (Sitzmann et al., 2020b) and many more. COIN-like approaches that convert data to INRs and compress these are therefore promising for building flexible neural codecs applicable to a range of modalities.

In this paper, we identify and address several key problems with COIN and propose a compression algorithm applicable to multiple modalities, which we call COIN++. More specifically, we identify the following issues with COIN: *1.* Encoding is slow: compressing a single image can take up to an hour, *2.* Lack of shared structure: as each image is compressed independently, there is no shared information between networks, *3.* Performance is well below state of the art (SOTA) image codecs. We address these issues by: *1.* Using meta-learning to reduce encoding time by more than two orders of magnitude to less than a second, compared to minutes or hours for COIN, *2.* Learning a base network that encodes shared structure and applying modulations to this network to encode instance specific information, *3.* Quantizing and entropy coding the modulations. While our method significantly exceeds COIN both in terms of compression and speed, it only partially closes the gap to SOTA codecs on well-studied modalities such as images. However, COIN++ is applicable to a wide range of data modalities where traditional methods are difficult to use, making it a promising tool for neural compression in non-standard domains.

## 2 Method

In this paper, we consider compressing data that can be expressed in terms of sets of coordinates $\mathbf{x} \in \mathcal{X}$ and features $\mathbf{y} \in \mathcal{Y}$. An image for example can be described by a set of pixel locations $\mathbf{x} = (x, y)$ in $\mathbb{R}^2$ and their corresponding RGB values $\mathbf{y} = (r, g, b)$ in $\{0, 1, ..., 255\}^3$. Similarly, an MRI scan can be described by a set of positions in 3D space $\mathbf{x} = (x, y, z)$ and an intensity value $\mathbf{y} \in \mathbb{R}^+$. Given a single datapoint as a collection of coordinate and feature pairs $\mathbf{d} = \{(\mathbf{x}_i, \mathbf{y}_i)\}_{i=1}^n$ (for example an image as a collection of $n$ pixel locations and RGB values), the COIN approach consists in fitting a neural network $f_\theta : \mathcal{X} \to \mathcal{Y}$ with parameters $\theta$ to the datapoint by minimizing

$$\mathcal{L}(\theta, \mathbf{d}) = \sum_{i=1}^n \|f_\theta(\mathbf{x}_i) - \mathbf{y}_i\|_2. \tag{1}$$

The weights $\theta$ are then quantized and stored as a compressed representation of the datapoint $\mathbf{d}$. The neural network $f_\theta$ is parameterized by a SIREN (Sitzmann et al., 2020b), i.e. an MLP with sine activation functions, which is necessary to fit high frequency data such as natural images (Mildenhall et al., 2020; Tancik et al., 2020b; Sitzmann et al., 2020b). More specifically, a SIREN layer is defined by an elementwise sin applied to a hidden feature vector $\mathbf{h} \in \mathbb{R}^d$ as

$$\text{SIREN}(\mathbf{h}) = \sin(\omega_0(W\mathbf{h} + \mathbf{b})) \tag{2}$$

where $W \in \mathbb{R}^{d \times d}$ is a weight matrix, $\mathbf{b} \in \mathbb{R}^d$ a bias vector and $\omega_0 \in \mathbb{R}^+$ a positive scaling factor.

While this approach is very general, there are several key issues. Firstly, as compression involves minimizing equation 1, encoding is extremely slow. For example, compressing a single image from the Kodak dataset (Kodak, 1991) takes nearly an hour on a 1080Ti GPU (Dupont et al., 2021a). Secondly, as each datapoint $\mathbf{d}$ is fitted with a separate neural network $f_\theta$, there is no information shared across datapoints. This is clearly suboptimal when several datapoints are available: natural images for example share a lot of common structure that does not need to be repeatedly stored for each individual image. In the following sections, we show how our proposed approach, COIN++, addresses these problems while maintaining the generality of COIN.

## 2.1 Storing modulations

While COIN stores each image as a separate neural network, we instead train a base network shared across datapoints and apply *modulations* to this network to parameterize individual datapoints. Given a base network, such as a multi-layer perceptron (MLP), we use FiLM layers (Perez et al., 2018), to modulate the hidden features $\mathbf{h} \in \mathbb{R}^d$ of the network by applying elementwise scales $\boldsymbol{\gamma} \in \mathbb{R}^d$ and shifts $\boldsymbol{\beta} \in \mathbb{R}^d$ as

$$\text{FiLM}(\mathbf{h}) = \boldsymbol{\gamma} \odot \mathbf{h} + \boldsymbol{\beta}. \tag{3}$$

Given a fixed base MLP, we can therefore parameterize families of neural networks by applying different scales and shifts at each layer. Each neural network function is therefore specified by a set of scales and shifts, which are collectively referred to as modulations (Perez et al., 2018). Recently, the FiLM approach has also been applied in the context of INRs. Chan et al. (2021) parameterize the generator in a generative adversarial network by a SIREN network and generate samples by applying modulations to this network as $\sin(\boldsymbol{\gamma} \odot (W\mathbf{h} + \mathbf{b}) + \boldsymbol{\beta})$. Similarly, Mehta et al. (2021) parameterize families of INRs using a scale factor via $\boldsymbol{\alpha} \odot \sin(W\mathbf{h} + \mathbf{b})$. Both of these approaches can be modified to use a low dimensional latent vector mapped to a set of modulations instead of directly applying modulations. Chan et al. (2021) map a latent vector to scales and shifts with an MLP, while Mehta et al. (2021) map the latent vector through an MLP of the same shape as the base network and use the hidden activations of this network as modulations. However, we found that both of these approaches performed poorly in terms of compressibility, requiring a large number of modulations to achieve satisfying reconstructions.

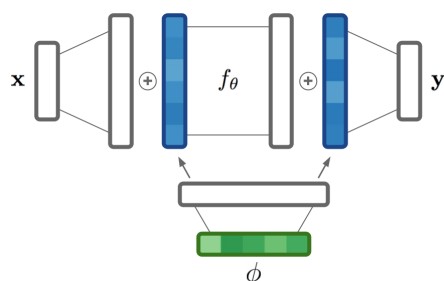

Figure 2: COIN++ architecture. Latent modulations $\phi$ (in green) are mapped through a hypernetwork to modulations (in blue) which are added to activations of the base network $f_\theta$ (in white) to parameterize a single function that can be evaluated at coordinates $\mathbf{x}$ to obtain features $\mathbf{y}$.

Instead, we propose a new parameterization of modulations for INRs which, on top of yielding better compression rates, is also more stable to train. More specifically, given a base SIREN network, we only apply shifts $\boldsymbol{\beta} \in \mathbb{R}^d$ as modulations using

$$\sin(\omega_0(W\mathbf{h} + \mathbf{b} + \boldsymbol{\beta})) \tag{4}$$

at every layer of the MLP. To further reduce storage, we use a latent vector which is linearly mapped to the modulations as shown in Figure 2. In a slight overload of notation, we also refer to this vector as modulations or latent modulations. Indeed, we found empirically that using only shifts gave the same performance as using both shifts and scales while using only scales yielded considerably worse performance. In addition, linearly mapping the latent vector to modulations worked better than using a deep MLP as in Chan et al. (2021). Given this parameterization, we then store a datapoint d (such as an image) as a set of (latent) modulations $\phi$. To decode the datapoint, we simply evaluate the modulated base network $f_\theta(\cdot; \phi)$ at every coordinate $\mathbf{x}$,

$$\mathbf{y} = f_\theta(\mathbf{x}; \phi) \tag{5}$$

as shown in Figure 3. To fit a set of modulations $\phi$ to a datapoint $\mathbf{d}$, we keep the parameters $\theta$ of the

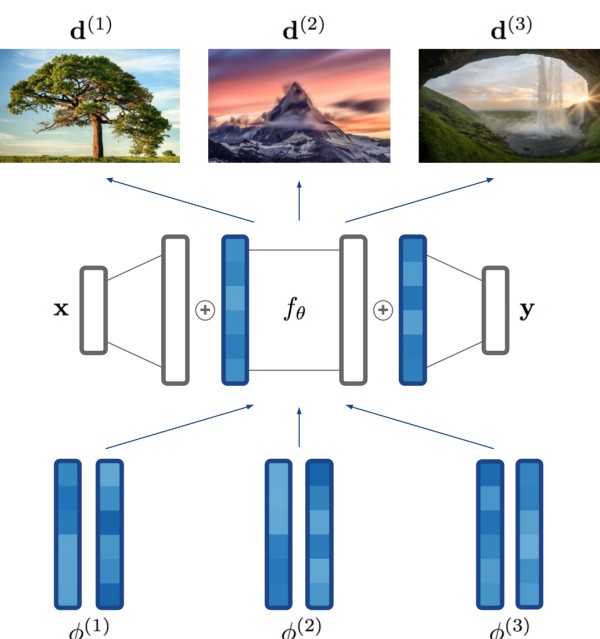

Figure 3: By applying modulations $\phi^{(1)}$, $\phi^{(2)}$, $\phi^{(3)}$ to a base network $f_\theta$, we obtain different functions that can be decoded into datapoints $\mathbf{d}^{(1)}$, $\mathbf{d}^{(2)}$, $\mathbf{d}^{(3)}$ by evaluating the functions at various coordinates. While we show images in this figure, the same principle can be applied to a range of data modalities.

base network fixed and minimize

$$\mathcal{L}(\theta, \phi, \mathbf{d}) = \sum_{i=1}^{n} \|f_\theta(\mathbf{x}_i; \phi) - \mathbf{y}_i\|_2 \qquad (6)$$

over $\phi$. In contrast to COIN, where each datapoint $\mathbf{d}$ is stored as a separate neural network $f_\theta$, COIN++ only requires storing $O(k)$ modulations (or less when using latents) as opposed to $O(k^2)$ weights, where $k$ is the width of the MLP. In addition, this approach allows us to store shared information in the base network and instance specific information in the modulations. For natural images for example, the base network encodes structure that is common to natural images while the modulations store the information required to reconstruct individual images.

## 2.2 Meta-learning modulations

Given a base network $f_\theta$, we can encode a datapoint $\mathbf{d}$ by minimizing equation 6. However, we are still faced with two problems: *1.* We need to learn the weights $\theta$ of the base network, *2.* Encoding a datapoint via equation 6 is slow, requiring thousands of iterations of gradient descent. COIN++ solves both of these problems with meta-learning.

Recently, Sitzmann et al. (2020a); Tancik et al. (2020a) have shown that applying Model-Agnostic Meta-Learning (MAML) (Finn et al., 2017) to INRs can reduce fitting at test time to just a few gradient steps. Instead of minimizing $\mathcal{L}(\theta, \mathbf{d})$ directly via gradient descent from a random initialization, we can meta-learn an initialization $\theta^*$ such that minimizing $\mathcal{L}(\theta, \mathbf{d})$ can be done in a few gradient steps. More specifically, assume we are given a dataset of $N$ points $\{\mathbf{d}^{(j)}\}_{j=1}^N$. Starting from an initialization $\theta$, a step of the MAML inner loop on a datapoint $\mathbf{d}^{(j)}$ is given by

$$\theta^{(j)} = \theta - \alpha\nabla_\theta\mathcal{L}(\theta, \mathbf{d}^{(j)}), \qquad (7)$$

where $\alpha$ is the inner loop learning rate. We are then interested in learning a good initialization $\theta^*$ such that the loss $\mathcal{L}(\theta, \mathbf{d}^{(j)})$ is minimized after a few gradient steps across the entire set of datapoints $\{\mathbf{d}^{(j)}\}_{j=1}^N$. To update the initalization $\theta$, we then perform a step of the outer loop, with an outer loop learning rate $\beta$, via

$$\theta \leftarrow \theta - \beta\nabla_\theta \sum_{j=1}^N \mathcal{L}(\theta^{(j)}, \mathbf{d}^{(j)}). \qquad (8)$$

In our case, MAML cannot be used directly since at test time we only fit the modulations $\phi$ and not the shared parameters $\theta$. We therefore need to meta-learn an initialization for $\theta$ and $\phi$ such that, given a new datapoint, the *modulations* $\phi$ can rapidly be computed while keeping $\theta$ constant. Indeed, we only store the modulations for each datapoint and share the parameters $\theta$ across all datapoints. For COIN++, a single step of the inner loop is then given by

$$\phi^{(j)} = \phi - \alpha\nabla_\phi\mathcal{L}(\theta, \phi, \mathbf{d}^{(j)}), \qquad (9)$$

where $\theta$ is kept fixed. Performing the inner loop on a subset of parameters has previously been explored by Zintgraf et al. (2019) and is referred to as CAVIA. As observed in CAVIA, meta-learning the initialization for $\phi$ is redundant as it can be absorbed into a bias parameter of the base network weights $\theta$. We therefore only need to meta-learn the shared parameter initialization $\theta$. The update rule for the outer loop is then given by

$$\theta \leftarrow \theta - \beta\nabla_\theta \sum_{j=1}^N \mathcal{L}(\theta, \phi^{(j)}, \mathbf{d}^{(j)}). \qquad (10)$$

The inner loop then updates the modulations $\phi$ while the outer loop updates the shared parameters $\theta$. This algorithm allows us to meta-learn a base network such that each set of modulations can easily and rapidly be fitted (see Figure 4). In practice, we find that as few as 3 gradient steps gives us compelling results, compared with thousands for COIN.

## 2.3 Patches, quantization and entropy coding for modulations

**Patches for large scale data**. While meta-learning the base network allows us to rapidly encode new datapoints into modulations, the training procedure is expensive, as MAML must take gradients through the

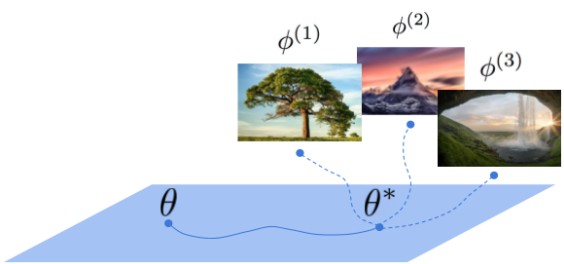 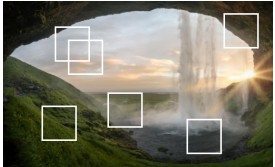 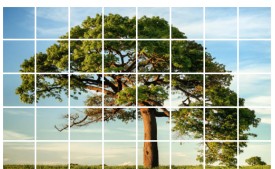

Figure 4: (Left) Starting from a random initialization $\theta$, we meta-learn parameters $\theta^*$ of the base network (with training progress shown as a solid line) such that modulations $\phi$ can easily be fit in a few gradient steps (with fitting shown in dashed lines). (Right) During training we sample patches randomly, while at test time we partition the datapoint into patches and fit modulations to each patch.

inner loop (Finn et al., 2017). For large datapoints (such as high resolution images or MRI scans), this can become prohibitively expensive. While first-order approximations exist (Finn et al., 2017; Nichol et al., 2018; Rajeswaran et al., 2019), we found that they severely hindered performance. Instead, to reduce memory usage, we split datapoints into random patches during training. For large scale images for example, we train on 32×32 patches. At train time, we then learn a base network such that modulations can easily be fit to patches. At test time, we split a new image into patches and compute modulations for each of them. The image is then represented by the set of modulations for all patches (see Figure 4). We use a similar approach for other data modalities, e.g. MRI scans are split into 3D patches.

**Quantization**. While COIN quantizes the neural network weights from 32 bits to 16 bits to reduce storage, quantizing beyond this severely hinders performance (Dupont et al., 2021a). In contrast, we find that modulations are surprisingly quantizable. During meta-learning, modulations are represented by 32 bit floats. To quantize these to shorter bitwidths, we simply use uniform quantization. We first clip the modulations to lie within 3 standard deviations of their mean. We then split this interval into $2^b$ equally sized bins (where $b$ is the number of bits). Remarkably, we found that reducing the number of bits from 32 to 5 (i.e. reducing the number of symbols from more than $10^9$ to only 32) resulted only in small decreases in reconstruction accuracy. Simply applying uniform quantization then improves compression by a factor of 6 at little cost in reconstruction quality.

**Entropy coding**. A core component of almost all codecs is entropy coding, which allows for lossless compression of the quantized code, using e.g. arithmetic coding (Rissanen & Langdon, 1979). This relies on a model of the distribution of the quantized codes. As with quantization, we use a very simple approach for modeling this distribution: we count the frequency of each quantized modulation value in the training set and use this distribution for arithmetic coding at test time. In our experiments, this reduced storage 8-15% at no cost in reconstruction quality. While this simple entropy coding scheme works well, we expect more sophisticated methods to significantly improve performance, which is an exciting direction for future work.

Finally, we note that we only transmit the *modulations* and assume the receiver has access to the shared base network. As such, only the modulations are quantized, entropy coded and count towards the final compressed file size. This is similar to the typical neural compression setting where the receiver is assumed to have access to the autoencoder and only the quantized and entropy coded latent vector is transmitted.

## 3 Related Work

**Neural compression**. Learned compression approaches are typically based on autoencoders that jointly minimize rate and distortion, as initially introduced in Ballé et al. (2017); Theis et al. (2017). Ballé et al. (2018) extend this by adding a hyperprior, while Mentzer et al. (2018); Minnen et al. (2018); Lee et al. (2019) use an autoregressive model to improve entropy coding. Cheng et al. (2020b) improve the accuracy of the entropy models by adding attention and Gaussian mixture models for the distribution of latent codes, while Xie et al. (2021) use invertible convolutional layers to further enhance performance. While most of these are optimized on traditional distortion metrics such as MSE or SSIM, other works have explored the use of

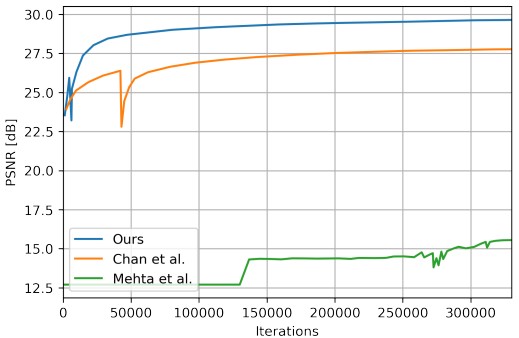 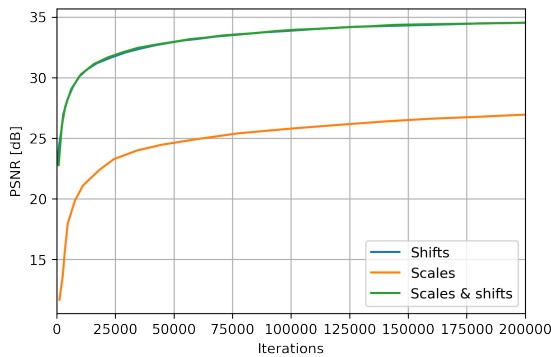

Figure 5: (Left) Test PSNR on CIFAR10 during training using a fixed number of modulations (see appendix for experimental details). Our method outperforms both baselines, improving PSNR by 2dB for the same number of parameters. (Right) Comparison of using shifts, scales and scales & shifts for modulations on MNIST (note that shifts and scales & shifts overlap). As can be seen, shifts perform significantly better than scales.

generative adversarial networks for optimizing perceptual metrics (Agustsson et al., 2019; Mentzer et al., 2020). Neural compression has also been applied to video (Lu et al., 2019; Goliński et al., 2020; Agustsson et al., 2020) and audio (Kleijn et al., 2018; Valin & Skoglund, 2019; Zeghidour et al., 2021).

**Implicit neural representations and compression**. In addition to COIN, several recent works have explored the use of INRs for compression. Davies et al. (2020) encode 3D shapes with neural networks and show that this can reduce memory usage compared with traditional decimated meshes. Chen et al. (2021) represent videos by convolutional neural networks that take as input a time index and output a frame in the video. By pruning, quantizing and entropy coding the weights of this network, the authors achieve compression performance close to standard video codecs. Lee et al. (2021) meta-learn sparse and parameter efficient initializations for INRs and show that this can reduce the number of parameters required to store an image at a given reconstruction quality, although it is not yet competitive with image codecs such as JPEG. Lu et al. (2021); Isik et al. (2021) explore the use of INRs for volumetric compression. Zhang et al. (2021) compress frames in videos using INRs (which are quantized and entropy coded) while learning a flow warping to model differences between frames. Results on video benchmarks are promising although the performance still lags behind standard video codecs. In concurrent work, Strümpler et al. (2021) propose a method for image compression with INRs which is closely related to ours. The authors also meta-learn an MLP initialization and subsequently quantize and entropy code the weights of MLPs fitted to images, leading to large performance gains over COIN. In particular, for large scale images their method significantly outperforms both COIN and COIN++ as they do not use patches. However, their approach still requires tens of thousands of iterations at test time to fully converge, unlike ours which requires 10 iterations (three orders of magnitude faster). Further, the authors do not employ modulations but directly learn the weights of the MLPs at test time. Finally, unlike our work, their approach is not applied to a wide range of modalities, including audio, medical and climate data. Indeed, to the best of our knowledge, none of these works have considered INRs for building a unified compression framework across data modalities.

# 4 Experiments

We evaluate COIN++ on four data modalities: images, audio, medical data and climate data. We implement all models in PyTorch (Paszke et al., 2019) and train on a single GPU. We use SGD for the inner loop with a learning rate of 1e-2 and Adam for the outer loop with a learning rate of 1e-6 or 3e-6. We normalize coordinates $\mathbf{x}$ to lie in $[-1, 1]$ and features $\mathbf{y}$ to lie in $[0, 1]$. Full experimental details required to reproduce all the results can be found in the appendix. We train COIN++ using MSE between the compressed and ground truth data. As is standard, we measure reconstruction performance (or distortion) using PSNR (in dB), which is defined as $\text{PSNR} = -10 \log_{10}(\text{MSE})$. We measure the size of the compressed data (or

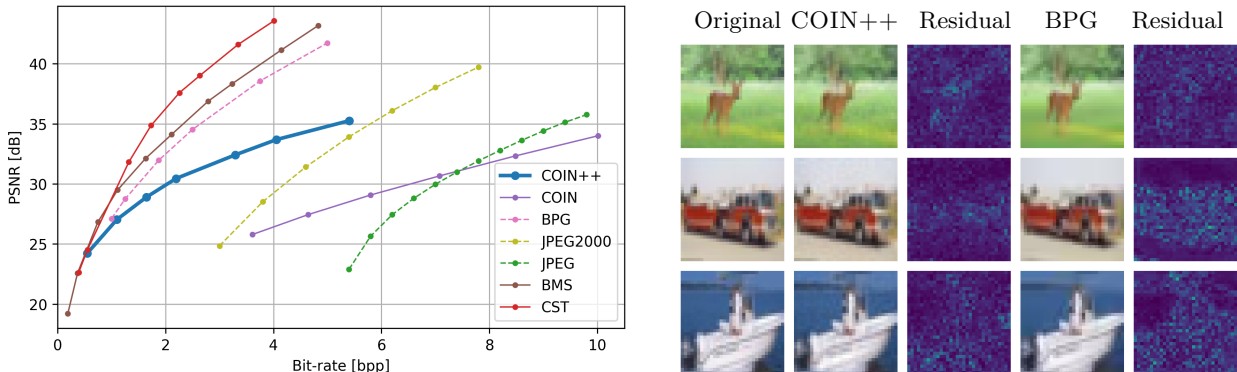

Figure 6: (Left) Rate distortion plot on CIFAR10. COIN++ outperforms COIN, JPEG and JPEG2000 while partially closing the gap to state of the art codecs. (Right) Qualitative comparison of compression artifacts for models at similar reconstruction quality. COIN++ achieves 32.4dB at 3.29 bpp while BPG achieves 31.9dB at 1.88 bpp.

rate) in terms of bits-per-pixel (bpp) which is given by $\frac{\text{number of bits}}{\text{number of pixels}}$[1] and kilobits per second (kpbs) for audio. We benchmark COIN++ against a large number of baselines including standard image codecs - JPEG (Wallace, 1992), JPEG2000 (Skodras et al., 2001), BPG (Bellard, 2014) and VTM (Bross et al., 2021) - autoencoder based neural compression - BMS (Ballé et al., 2018), MBT (Minnen et al., 2018) and CST (Cheng et al., 2020b) - standard audio codecs - MP3 (MP3, 1993) - and COIN (Dupont et al., 2021a). For clarity, we use consistent colors for different codecs and plot learned codecs with solid lines and standard codecs with dashed lines. The code to reproduce all experiments in the paper can be found at https://github.com/EmilienDupont/coinpp.

## 4.1 Comparisons to other INR parameterizations

We first compare our parameterization of INRs with the methods proposed by Chan et al. (2021) and Mehta et al. (2021) as described in Section 2.1. As can be seen in Figure 5, our method significantly outperforms both in terms of compressibility, improving PSNR by 2dB with the same number of parameters[2]. Further, using shift modulations is more effective than using scales and shifts, and performs significantly better than scales alone. We also note that our method allows us to quickly fit INRs using only a few hundred parameters. This is in contrast to existing works on meta-learning for INRs (Sitzmann et al., 2020a; Tancik et al., 2020a), which typically require fitting 3 orders of magnitude more parameters at test time.

## 4.2 Images: CIFAR10

We train COIN++ on CIFAR10 using 128, 256, 384, 512, 768 and 1024 latent modulations. As can be seen in Figure 6, COIN++ vastly outperforms COIN, JPEG and JPEG2000 while partially closing the gap to BPG, particularly at low bitrates. To the best of our knowledge, this is the first time compression with INRs has outperformed image codecs like JPEG2000. Part of the gap between COIN++ and SOTA codecs (BMS, CST) is likely due to entropy coding: we use the simple scheme described in Section 2.3, while BMS and CST use deep generative models. We hypothesize that using deep entropy coding for the modulations would significantly reduce this gap. Figure 6 shows qualitative comparisons between our model and BPG to highlight the types of compression artifacts obtained with COIN++. In order to thoroughly analyse and evaluate each component of COIN++, we perform a number of ablation studies.

**Quantization bitwidth**. Quantizing the modulations to a lower bitwidth yields more compressed codes at the cost of reconstruction accuracy. To understand the tradeoff between these, we show rate distortion plots when quantizing from 3 to 8 bits in Figure 7a. As can be seen, the optimal bitwidths are surprisingly low:

---

[1]For non image data a "pixel" corresponds to a single dimension of the data.
[2]Despite significant experimental effort, we were unable to achieve better performance using meta-learning with Mehta et al. (2021).

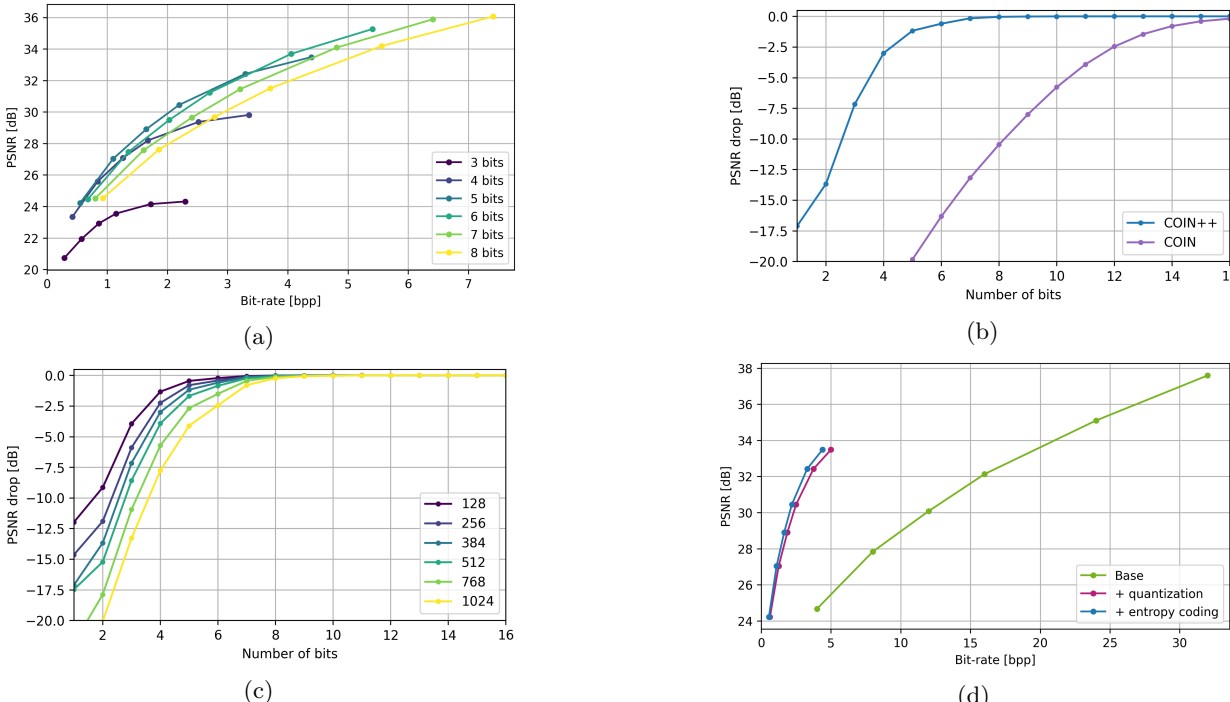

Figure 7: (a) Rate distortion plot on CIFAR10 when quantizing the modulations $\phi$ to various bitwidths. As can be seen, using 5 or 6 bits is optimal. (b) Drop in PSNR for COIN and COIN++ quantization. COIN++ is significantly more robust to quantization. (c) Drop in PSNR when quantizing the modulations $\phi$ to various bitwidths, for various latent dimensions. Larger latent dimensions have larger drops in PSNR. (d) Effect of of quantization (to 5 bits) and entropy coding on CIFAR10. As can be seen, both quantization and entropy coding improve rate distortion performance.

5 bits is optimal at low bitrates while 6 is optimal at higher bitrates. Qualitative artifacts obtained from quantizing the modulations are shown in Figure 14 in the appendix.

**Quantization COINvs COIN++**. We compare the drop in PSNR due to quantization for COIN and COIN++ in Figure 7b. As can be seen, modulations are remarkably quantizable: when quantizing the COIN weights directly, performance decreases significantly around 14 bits, whereas quantizing modulations yields small drops in PSNR even when using 5 bits. However, as shown in Figure 7c, the drop in PSNR from quantization is larger for larger models.

**Entropy coding**. Figure 7d shows rate distortion plots for full precision, quantized and entropy coded modulations. As can be seen, both quantization and entropy coding significantly improve performance.

**Encoding/decoding speed**. Table 1 shows the average encoding and decoding time for BPG, COIN and COIN++ on CIFAR10. As BPG runs on CPU while COIN and COIN++ run on GPU, these times are not directly comparable. However, we follow standard practice in the literature and run all neural codecs on GPU and standard codecs on CPU (see appendix B.1 for hardware details). In terms of encoding, COIN++ compresses images $300\times$ faster than COIN while achieving a $4\times$ better compression rate. Note that these results are obtained from compressing each image separately. When using batches of images, we can compress the entire CIFAR10 test set (10k images) in 4mins when using 10 inner loop steps (and in just over a minute when using 3 steps). In addition, as shown in Figure 15 in the appendix, COIN++ requires only 3 gradient steps to reach the same performance as COIN does in 10,000 steps, while using $4\times$ less storage. In terms of decoding, COIN++ is slower than COIN as it uses a larger shared network and entropy coding. However, decoding with COIN++ remains fast (on the order of a millisecond).

|  | BPG | COIN | COIN++ |
|---|---|---|---|
| Encoding (ms) | 5.19 | $2.97 \times 10^4$ | 94.9 |
| Decoding (ms) | 1.25 | 0.46 | 1.29 |

Table 1: Average encoding and decoding time on CIFAR10 for BPG, COIN and COIN++. As can be seen, COIN++ encodes images orders of magnitude faster than COIN, while only being marginally slower than BPG at decoding time.

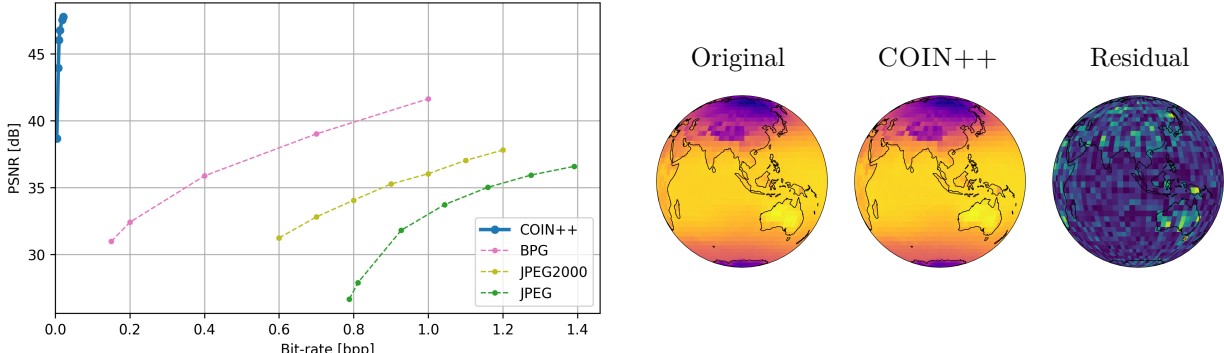

Figure 8: (Left) Rate distortion plot on ERA5. COIN++ vastly outperforms all baselines. (Right) COIN++ compression artifacts on ERA5. See appendix E.5 for more samples.

### 4.3 Climate data: ERA5 global temperature measurements

To demonstrate the flexibility of our approach, we also use COIN++ to compress data lying on a manifold. We use global temperature measurements from the ERA5 dataset (Hersbach et al., 2019) with the processing and splits from Dupont et al. (2021b). The dataset contains 8510 train and 2420 test globes of size $46 \times 90$, with temperature measurements at equally spaced latitudes $\lambda$ and longitudes $\varphi$ on the Earth from 1979 to 2020. To model this data, we follow Dupont et al. (2021b) and use spherical coordinates $\mathbf{x} = (\cos \lambda \cos \varphi, \cos \lambda \sin \varphi, \sin \lambda)$ for the inputs. As a baseline, we compare COIN++ against JPEG, JPEG2000 and BPG applied to flat map projections of the data. As can be seen in Figure 8, COIN++ vastly outperforms all baselines. These strong results highlight the versatility of the COIN++ approach: unlike traditional codecs and autoencoder based methods (which would require spherical convolutions for the encoder), we can easily apply our method to a wide range of data modalities, including data lying on a manifold. Indeed, COIN++ achieves a 3000× compression rate while having an RMSE of 0.5°C, highlighting the potential for compressing climate data. More generally, it is likely that many highly compressible data modalities are not compressed in practice, simply because an applicable codec does not exist. We hope the flexibility of COIN++ will help make neural compression more generally applicable to such modalities.

We also note that very few baselines exist for compressing data on manifolds. A notable exception is McEwen et al. (2011) which builds a codec analogous to JPEG2000 by using wavelet transforms on the sphere. However, their method is not likely to outperform ours as it is not a learned codec and so cannot take advantage of the low entropy of the climate data. Further, their method is only applicable to the sphere, while our method is applicable to any manifold where a coordinate system can be defined to pass as input to the INR. Finally, we note that, while autoencoder based methods may be able to outperform COIN++ on this data modality, building such an autoencoder would require a non-trivial amount of work (including the use of spherical convolutions both for the encoder/decoder and hyperprior). In contrast, with COIN++ we simply change the input coordinates.

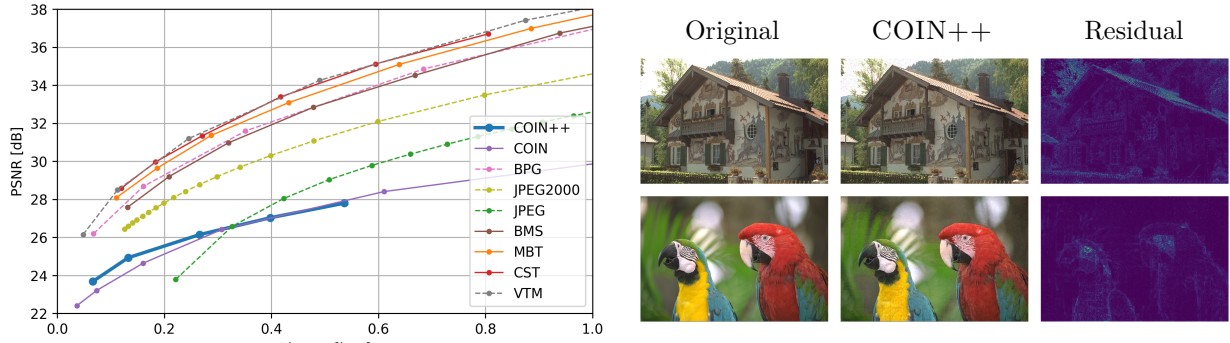

Figure 9: (Left) Rate distortion plot on Kodak. While COIN++ performs slightly better than COIN, the use of patches reduces compression performance. (Right) COIN++ compression artifacts on Kodak. See appendix E.5 for more samples.

## 4.4 Compression with patches

To evaluate the patching approach from Section 2.3 and to demonstrate that COIN++ can scale to large data (albeit at a cost in performance), we test our model on images, audio and MRI data.

**Large scale images: Kodak**. The Kodak dataset (Kodak, 1991) contains 24 large scale images of size 768×512. To train the model, we use random 32×32 patches from the Vimeo90k dataset (Xue et al., 2019), containing 154k images of size 448×256. At evaluation time, each Kodak image is then split into 384 32×32 patches which are compressed independently. As we do not model the global structure of the image, we therefore expect a significant drop in performance compared to the case when no patching is required. As can be seen in Figure 9, the performance of COIN++ indeed drops, but still outperforms COIN and JPEG at low bitrates. We expect that this can be massively improved by modeling the global structure of the image (e.g. two patches of blue sky are nearly identical, but that information redundancy is not exploited in the current setup) but leave this to future work.

**Audio: LibriSpeech**. To evaluate COIN++ on audio, we use the LibriSpeech dataset (Panayotov et al., 2015) containing several hours of speech data recorded at 16kHz. As a baseline, we compare against the widely used MP3 codec (MP3, 1993). We split each audio sample into patches of varying size and compress each of these to obtain models at various bit-rates (we refer to appendix B.5 for full experimental details). As can be seen in Figure 10, even though audio is a very different modality from the rest considered in this paper, COIN++ can still be used for compression, highlighting the versatility of our approach. However, in terms of performance, COIN++ lags behind well-establised audio codecs such as MP3.

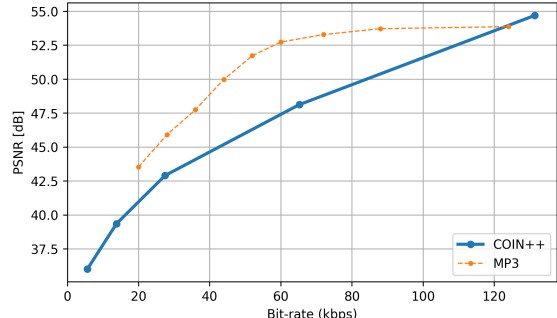

Figure 10: Rate distortion plot on LibriSpeech. While COIN++ does not outperform specialized codecs like MP3, it still performs fairly well on audio, even though this is a vastly different data modality.

**Medical data: brain MRI scans**. Finally, we train our model on brain MRI scans from the FastMRI dataset (Zbontar et al., 2018). The dataset contains 565 train volumes and 212 test volumes with sizes ranging from 16×320×240 to 16×384×384 (see appendix A.2 for full dataset details). As a baseline, we compare our model against JPEG, JPEG2000 and BPG applied independently to each slice. Due to memory constraints, we train COIN++ on 16×16×16 patches. We therefore store roughly 400 independent patches at test time (as opposed to 16 slices for the image codecs). Even then COIN++ performs reasonably well, particularly at low bitrates (see Figure 11). As a large number of patches are nearly identical, especially close to the edges, we expect that large gains can be made from modeling the global structure of the data. Qualitatively, our model also performs well although it has patch artifacts at low bitrates (see Figure 11).

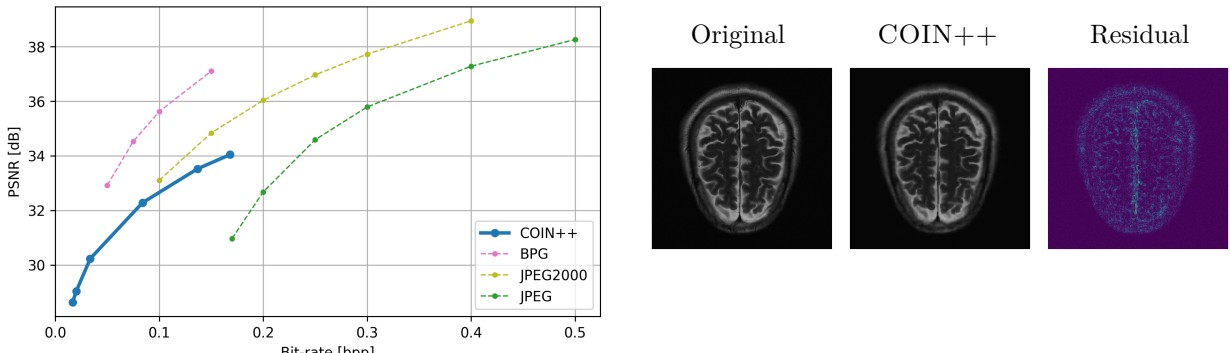

Figure 11: (Left) Rate distortion plot on FastMRI. As can be seen, even when using patches, COIN++ performs better than JPEG. However, our methods lags behind BPG and JPEG2000. (Right) COIN++ compression artifacts on FastMRI. See appendix E.5 for more samples.

## 5 Conclusion, limitations and future work

**Conclusion**. We introduce COIN++, the first (to the best of our knowledge) neural codec applied to a wide range of data modalities. Our framework significantly improves performance compared to COIN both in terms of compression and encoding time, while being competitive with well-established codecs such as JPEG. While COIN++ does not match the performance of SOTA codecs, we hope our work will help expand the range of domains where neural compression is applicable.

**Limitations**. The main drawback of COIN++ is that, because of the second-order gradients required for MAML, training the model is memory intensive. This in turn limits scalability and requires us to use patches for large data. Devising effective first-order approximations or bypassing meta-learning altogether would mitigate these issues. In addition, training COIN++ can occasionally be unstable, although the model typically recovers from loss instabilities (see Figure 12 in the appendix). Further, there are several common modalities our framework cannot handle, such as text or tabular data, as these are not easily expressible as continuous functions. Finally, COIN++ lags behind SOTA codecs. Indeed, while we have demonstrated the ease with which COIN++ is *applicable* to different modalities compared to autoencoders (as they do not require specialized and potentially complicated modality specific encoders and decoders), it is not clear whether COIN++ can *outperform* autoencoders specialized to each modality. However, we believe there are several interesting directions for future work to improve the performance of INR based codecs.

**Future work**. In its current form, COIN++ employs very basic methods for both quantization and entropy coding - using more sophisticated techniques for these two steps could likely lead to large performance gains. Indeed, recent success in modeling distributions of functions (Schwarz et al., 2020; Anokhin et al., 2021; Skorokhodov et al., 2021; Dupont et al., 2021b) suggests that large gains could be made from using deep generative models to learn the distribution of modulations for entropy coding. Similarly, better post-training quantization (Nagel et al., 2019; Li et al., 2021a) or quantization-aware training (Krishnamoorthi, 2018; Esser et al., 2020) would also improve performance. More generally, there are a plethora of methods from the model compression literature that could be applied to COIN++ (Cheng et al., 2020a; Liang et al., 2021). For large scale data, it would be interesting to model the global structure of patches instead of encoding and entropy coding them independently. Further, the field of INRs is progressing rapidly and these advances are likely to improve COIN++ too. For example, Martel et al. (2021) use adaptive patches to scale INRs to gigapixel images - such a partition of the input is similar to the variable size blocks used in BPG (Bellard, 2014). In addition, using better activation functions (Ramasinghe & Lucey, 2021) to increase PSNR and equilibrium models (Huang et al., 2021) to reduce memory usage are exciting avenues for future research.

Finally, as COIN++ replaces the encoder in traditional neural compression with a flexible optimization procedure and the decoder with a powerful functional representation, we believe compression with INRs has great potential. Advances in INRs, combined with more sophisticated entropy coding and quantization may

allow COIN-like algorithms to equal or even surpass SOTA codecs, while potentially allowing for compression on currently unexplored modalities.

**Acknowledgments**

We would like to thank Hyunjik Kim, Danilo Rezende, Dan Rosenbaum and Ali Eslami for helpful discussions. We thank Jean-Francois Ton for helpful discussions around meta-learning and for reviewing an early version of the paper. We also thank the anonymous reviewers for their valuable feedback which helped improve the paper. Emilien gratefully acknowledges his PhD funding from Google DeepMind.

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

## A   Dataset details

### A.1   Vimeo90k

We use the Vimeo90k triplet dataset Xue et al. (2019) containing 73,171 3-frame sequences from videos at a resolution of 448×256. We processed the dataset following Bégaint et al. (2020). The resulting dataset contains 153,939 training images and 11,346 test images.

### A.2   FastMRI

To generate the dataset, we use the validation split from the FastMRI brain multicoil database (Zbontar et al., 2018). This contains 1378 fully sampled brain MRI images obtained through a variety of sources - T1, T1 post-contrast, T2 and FLAIR images. We then filter the dataset to only use scans from the T2 source. In addition, as the vast majority of volumes have 16 slices, we also filter by volumes with 16 slices. We then randomly split the filtered scans into a 565 training volumes and 212 testing volumes. The train dataset contains the following shapes (with their counts):

(16, 384, 384): 329

(16, 320, 320): 229

(16, 384, 312): 2

(16, 320, 260): 2

(16, 320, 240): 1

(16, 384, 342): 1

(16, 320, 270): 1

While the test dataset contains the following shapes (with their counts):

(16, 384, 384): 124

(16, 320, 320): 86

(16, 320, 260): 2

We also normalize the data to lie in $[0, 1]$ (while COIN++ can handle data in any range, we cannot apply the image compression baselines if the data is not in $[0, 1]$). As the data contains outliers, we first compute a histogram of the data distribution and choose the maximum value such that 99.99% of the data has value less than this. We then normalize by the minimum and maximum value and clip any value lying outside this range (<0.01% of the data).

Disclaimer required when using the FastMRI dataset: *"Data used in the preparation of this article were obtained from the NYU fastMRI Initiative database (fastmri.med.nyu.edu) (Zbontar et al., 2018; Knoll et al., 2020). As such, NYU fastMRI investigators provided data but did not participate in analysis or writing of this report. A listing of NYU fastMRI investigators, subject to updates, can be found at:fastmri.med.nyu.edu. The primary goal of fastMRI is to test whether machine learning can aid in the reconstruction of medical images."*

### A.3   ERA5

The climate dataset was extracted from the ERA5 database (Hersbach et al., 2019), using the processing and splits from Dupont et al. (2021b) (see this reference for details). The resulting dataset contains 12,096 grids of size 46×90, with 8510 training examples, 1166 validation examples and 2420 test examples.

### A.4   LibriSpeech

The LibriSpeech dataset (Panayotov et al., 2015) contains several hours of read English Speech recorded at 16kHz. For training, we use the train-clean-100 split containing 28,539 examples and the test-clean split

containing 2,620 examples. We train and evaluate on the first 3 seconds of every example, corresponding to 48,000 audio samples per example.

## B  Experimental details

### B.1  CIFAR10

For all models, we set $\omega_0 = 50$ and used an inner learning rate of 1e-2, an outer learning rate of 3e-6 and batch size 64. All models were trained for 500 epochs (400k iterations). We used the following architectures:

- latent dim: 128, 10 layers of width 512
- latent dim: 256, 10 layers of width 512
- latent dim: 384, 10 layers of width 512
- latent dim: 512, 15 layers of width 512
- latent dim: 768, 15 layers of width 512
- latent dim: 1024, 15 layers of width 512

We used 10 inner steps at test time for all models.

While we only transmit modulations (which have between 128 and 1024 parameters), the receiver is assumed to have access to the based shared network, which has on the order of millions of parameters. For example, the latent dimension 384 model has 3.8 million parameters, corresponding to a model size of 15.5MB. We note that autoencoder based models typically have a similar size (Dupont et al., 2021a).

**INR baselines**. We used a latent dimension of 384 for all models. For Chan et al. (2021), we used a ReLU MLP with a single hidden layer with 512 units to map the latent vector to the modulations (we found that more layers performed worse). For Mehta et al. (2021), we used the exact model proposed in their paper and set the dimension of the **z** vector to 384.

**COINbaseline**. We manually searched for the best architecture for each bpp level. We followed all other hyperparameters from COIN (Dupont et al., 2021a) and trained for 10k iterations (we found this was enough to converge on CIFAR10). Surprisingly, we found that for CIFAR10 depth did not improve performance and that increasing the width of the layers was better. This may be because the layers are already very small.

- bpp: 3.6, 2 layers of width 12
- bpp: 4.6, 2 layers of width 14
- bpp: 5.8, 2 layers of width 16
- bpp: 7.1, 2 layers of width 18
- bpp: 8.5, 2 layers of width 20
- bpp: 10.0, 2 layers of width 22

For the COIN quantization experiments, we used uniform quantization for the weights and biases separately. We chose the number of standard deviations $k$ at which to define the quantization range using the formula $k = 3 + 3\frac{\text{number of bits} - 1}{15}$. I.e. when using 1 bit, we use 3 standard deviations and when using 16 bits we use 6 standard deviations. Indeed, there is a tradeoff between how much data we are cutting off and how finely we can quantize the range. We found that this formula generally gave robust results across different bit values.

**Autoencoder baselines**. All autoencoder baselines were trained using the CompressAI implementations (Bégaint et al., 2020). In order for these models to handle $32{\times}32$ images from the CIFAR10 dataset, we

modified the architectures both for BMS and CST. Specifically, for BMS we changed the last two convolutional layers in the encoder from kernel size 5, stride 2 convolutions to kernel size 3 stride 1 convolutions, in order to preserve the spatial size (we made similar changes for the transposed convolutions in the decoder). For CST we replaced the first three residual blocks in the image encoder with stride 1 convolutions instead of stride 2, hence preserving the size of the image. Similarly, we replaced the upsampling operations in the decoder with stride 1 upsampling (i.e. dimension preserving convolutions) instead of stride 2. Otherwise, we used the default parameters provided by CompressAI, i.e. for BMS, we used N=128 and M=192 and for CST N=128. We trained all models for 500 epochs with a learning rate of 1e-4. We trained models for each of the following $\lambda$ values: [0.0016, 0.0032, 0.0075, 0.015, 0.03, 0.05, 0.1, 0.15, 0.3, 0.5]. As particularly CST could be unstable to train, we trained two models for each value of $\lambda$ and kept the best model for the rate distortion plot.

**Standard image codec baselines**. We use three image codec baselines: JPEG (Wallace, 1992), JPEG2000 (Skodras et al., 2001) and BPG (Bellard, 2014). For each of these, we perform a search over either the quality, quantization level or compression ratio to find the best quality image (in terms of PSNR) at a given bpp level.

We use the JPEG implementation from Pillow version 8.1.0. We use the OpenJPEG version 2.4.0 implementation of JPEG2000, calling the binary file with

```
opj_compress -i <in filepath> -r <compression ratio> -o <out filepath>.
```

We use BPG version 0.9.8, calling the binary file with

```
bpgenc -f 444 -q <quantization level> -o <out filepath> <in filepath>.
```

**Encoding and decoding time**. We measure the encoding time of COIN and COIN++ on a 1080Ti GPU and the decoding time on a 2080Ti GPU. For COIN we fit a separate neural network for each image in the CIFAR10 test set and report the average encoding time. For COIN++ we similarly fit modulations for each image in the test set and report the average encoding time. For BPG, we measured encoding time on an AMD Ryzen 5 3600 (12) at 3.600GHz with 32GB of RAM.

## B.2  Kodak and Vimeo90k

For all models, we set $\omega_0 = 50$ and used an inner learning rate of 1e-2, an outer learning rate of 1e-6 and batch size 64. All models were trained for 600 epochs (1.4 million iterations). We used the following architectures:

- latent dim: 16, 10 layers of width 512
- latent dim: 32, 10 layers of width 512
- latent dim: 64, 10 layers of width 512
- latent dim: 96, 10 layers of width 512
- latent dim: 128, 10 layers of width 512

We used 32×32 patches from the Vimeo90k dataset to train the model and evaluated on the full Kodak images. We used 3 inner steps for the latent dim 32 and 64 models and 10 inner steps for the latent dim 16, 96 and 128 models as this gave the best results. We quantized all modulations to 5 bits.

We experimented with different patch sizes (8×8, 16×16, 32×32, 64×64) but found that 32×32 was optimal in our case. Indeed while using larger patches should help compression performance, it also requires more memory and hence a smaller batch size when memory is limited. As we use GPUs with only 11GB of RAM, the batch size we had to use for 64×64 was prohibitively small, leading to unstable training and worse performance.

## B.3  FastMRI

For all models, we set $\omega_0 = 50$ and used an inner learning rate of 1e-2, an outer learning rate of 3e-6 and batch size 16. All models were trained for 32,000 epochs (1.1 million iterations). We used the following architectures:

- latent dim: 16, 10 layers of width 512

- latent dim: 32, 10 layers of width 512

- latent dim: 64, 10 layers of width 512

- latent dim: 128, 10 layers of width 512

We trained on $16 \times 16 \times 16$ patches and evaluated on the full volumes. We used 10 inner steps at encoding time as this gave the best results. On the rate distortion plot, the first two points are the latent dim 16 model, quantized to 5 and 6 bits, then the latent dim 32 model, quantized to 5 bits, then the latent dim 64 model quantized to 6 bits and finally the latent dim 128 model, quantized to 5 bits and 6 bits.

### B.4  ERA5

For all models, we set $\omega_0 = 50$ and used an inner learning rate of 1e-2, an outer learning rate of 3e-6 and batch size 32. All models were trained for 800 epochs (210k iterations). We used the following architectures:

- latent dim: 4, 10 layers of width 384

- latent dim: 8, 10 layers of width 384

- latent dim: 12, 10 layers of width 384

We used 3 inner steps at encoding time as this gave the best results. On the rate distortion plot, the first two points are the latent dim 4 and 8 models quantized to 5 bits, then the latent dim 8 model quantized to 6 and 7 bits and finally the latent dim 12 model quantized to 7 and 8 bits.

### B.5  LibriSpeech

For all models, we set $\omega_0 = 50$ and used an inner learning rate of 1e-2, an outer learning rate of 1e-6 and batch size 64. We further scaled the coordinates to lie in $[-5, 5]$ as we found this improved performance (similar observations were made by Sitzmann et al. (2020b)). All models were trained for 1000 epochs (445k iterations), except the latent dim 256 model which was trained for 2000 epochs (890k iterations). We used the following architectures:

- latent dim: 128, 10 layers of width 512, patch size 1600

- latent dim: 128, 10 layers of width 512, patch size 800

- latent dim: 128, 10 layers of width 512, patch size 400

- latent dim: 128, 10 layers of width 512, patch size 200

- latent dim: 256, 10 layers of width 512, patch size 200

We used 3 inner steps at encoding time. On the rate distortion plot, each point corresponds to one of the above models quantized to 5, 6, 6, 7 and 7 bits respectively.

**Audio codec baselines**. We use the MP3 implementation from LAME version 3.100, calling the binary file with

```
lame -b <bit rate> <in filepath> <out filepath>.
```

## C    Figure details

Figure 7b (COIN vs COIN++ quantization). The results in this figure are averaged across the entire CIFAR10 test set. We used COIN and COIN++ models that achieve roughly the same PSNR (30-31dB), corresponding to the bpp 7.1 model for COIN and the latent dim 384 model for COIN++.

Table 1 (Encoding time). The BPG model uses 1.25 bpp (PSNR: 28.7dB), the COIN++ model 1.14 bpp (PSNR: 28.9dB) and the COIN model 7.1 bpp (PSNR: 30.7dB).

Figure 9 (Kodak qualitative samples). The COIN++ model used for this plot has a bpp of 0.537 (latent dim 128).

Figure 11 (FastMRI qualitative samples). The COIN++ model used for this plot has a bpp of 0.168 (latent dim 128).

Figure 8 (ERA5 qualitative samples). The COIN++ model used for this plot has a bpp of 0.012 (latent dim 8).

Figure 14 (Qualitative quantization). This figure uses the COIN++ model with a latent dim of 768.

## D    Things we tried that didn't work

- As MAML is very memory intensive, we experimented with first-order approximations. We ran first-order MAML as described in Finn et al. (2017), but found that this severely hindered performance. Further, methods such as REPTILE (Nichol et al., 2018) are not applicable to our problem, as the weights updated in the inner and outer loop are not the same.

- Mehta et al. (2021) use a similar approach to us for fitting INRs (without meta-learning) by using overlapping patches in images. However, we found that using overlapping patches yielded a worse tradeoff between reconstruction accuracy and number of modulations and therefore used non-overlapping patches throughout.

- We experimented with using a deep MLP (and the architecture from Mehta et al. (2021)) for mapping the latent vector to modulations but found that this decreased performance. As MLPs are strictly more expressive than linear mappings, we hypothesize that this is due to optimization issues arising from the meta-learning. If the base network is learned without meta-learning, it is likely a deep MLP would improve performance over a linear mapping.

# E   Additional results

## E.1   Meta-learning curves

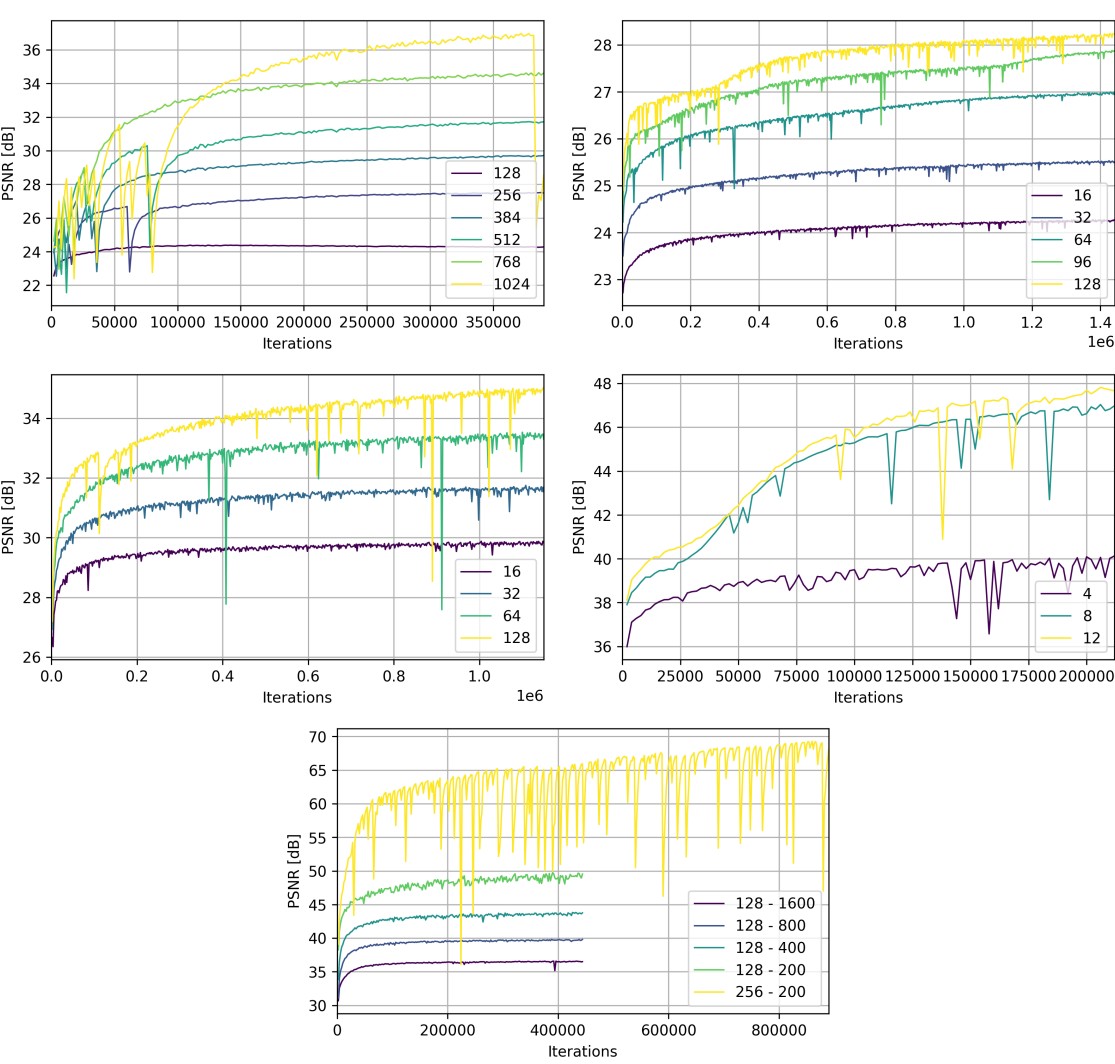

Figure 12: Validation PSNR (3 inner steps) during meta-learning on CIFAR10 (top left), Kodak (top right), FastMRI (middle left), ERA5 (middle right) and LibriSpeech (bottom). Note that for LibriSpeech, the legend corresponds to "latent dimension - patch size".

## E.2 CIFAR10 ablations

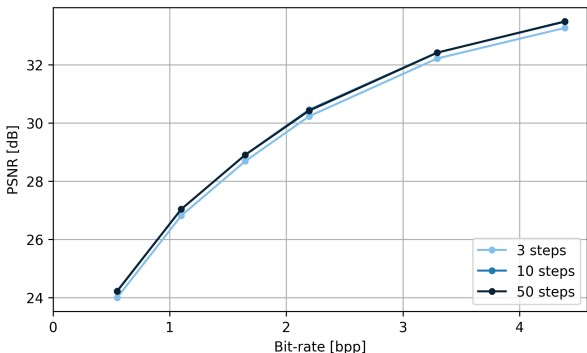

Figure 13: Effect of number of inner steps on CIFAR10 for a model that has been quantized to 5 bits, with entropy coding. While we use 3 inner steps for meta-learning, performing 10 steps at test time leads to an increase in reconstruction performance of 0.5-1.5dB, while fitting for more than 10 steps generally does not improve performance. Indeed, curves for 10 and 50 steps almost fully overlap.

## E.3 Qualitative quantization results

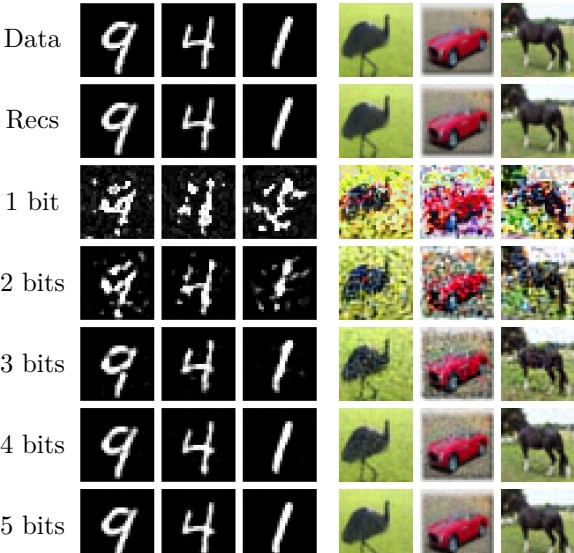

Figure 14: Qualitative effects of quantization. The top row shows ground truth data from MNIST and CIFAR10, the second row shows the reconstructions from full precision (32 bit) modulations. The subsequent rows show reconstructions when quantizing to various bitwidths. As can be seen, with only 5 bits, reconstructions are nearly perfect. Using as few as 1 or 2 bits, the class of the object is generally recognizable.

### E.4 Encoding curves

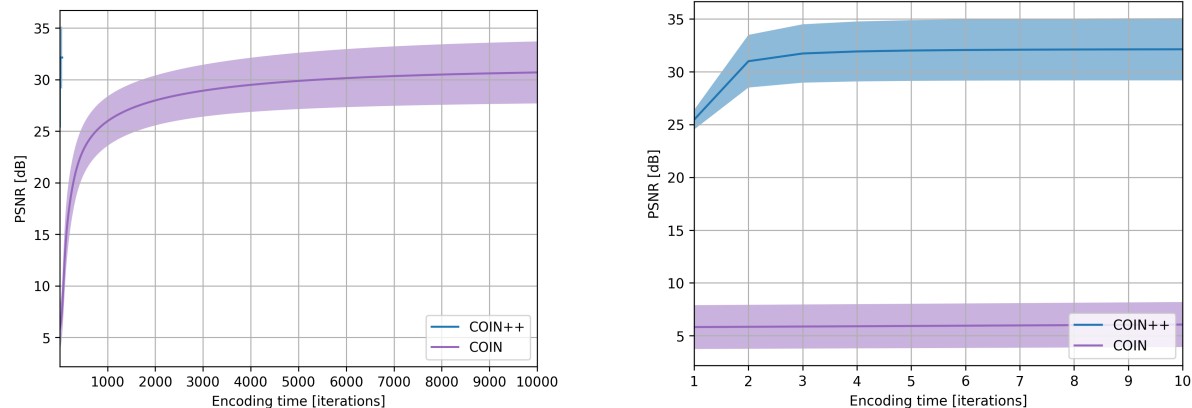

Figure 15: Encoding curves for COIN and COIN++ on CIFAR10 (full curve on the left, zoomed in version on the right). The COIN model has a bpp of 7.1, while COIN++ has a bpp of 2.2.

### E.5 Additional qualitative results

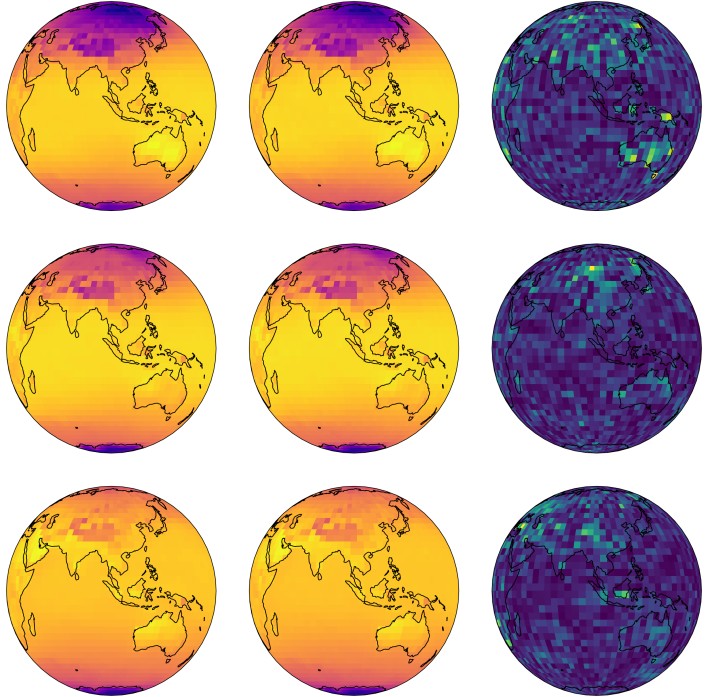

Figure 16: Qualitative compression artifacts on ERA5 using the latent dim 8 model with 0.012 bpp (original in first column, COIN++ in second column and residual in third column).

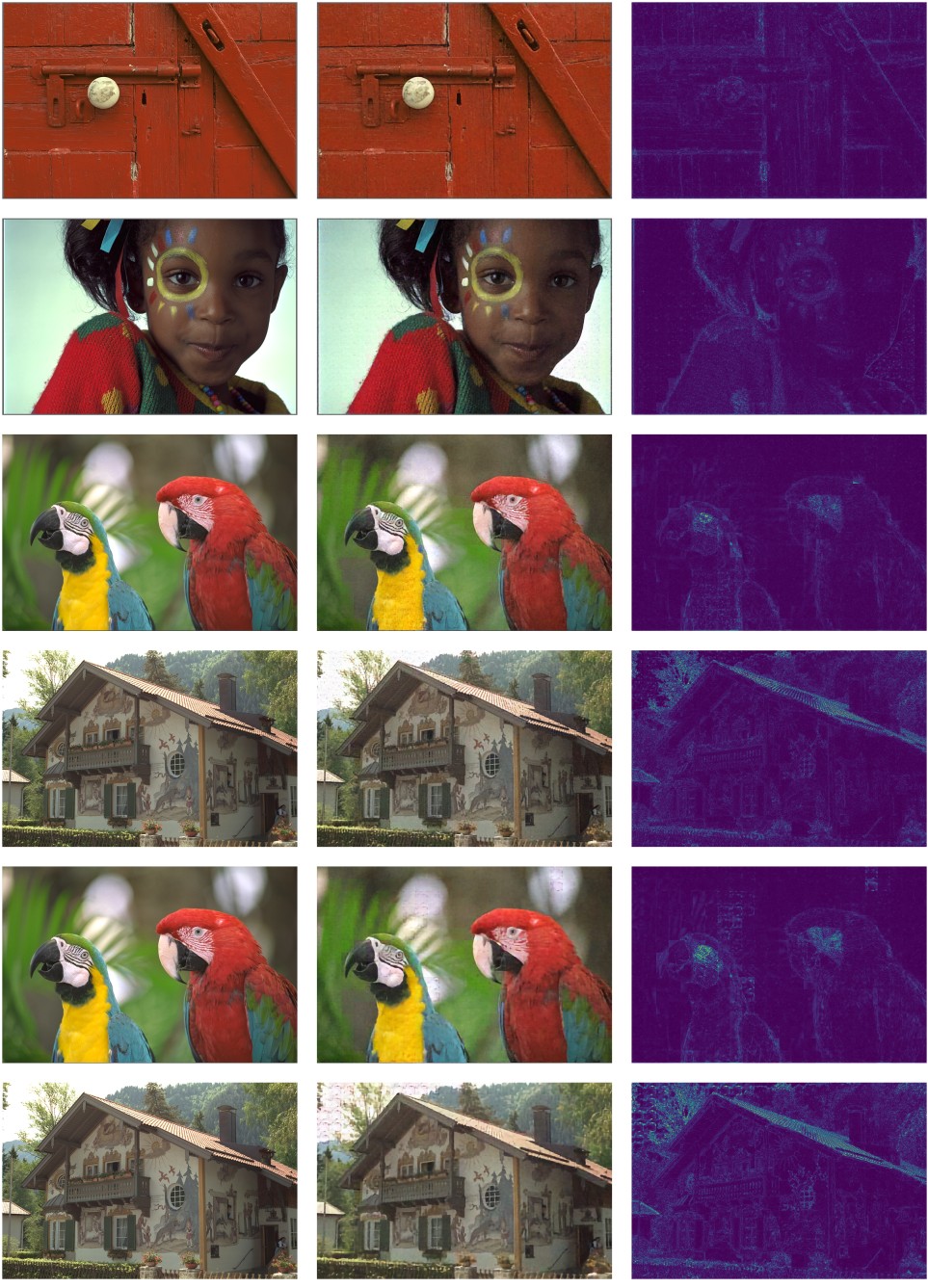

Figure 17: Qualitative compression artifacts on Kodak (original in first column, COIN++ in second column and residual in third column). The first 4 rows correspond to the model with latent dim 128 (0.537 bpp), while the bottom two rows correspond to the model with latent dim 64 (0.398 bpp).

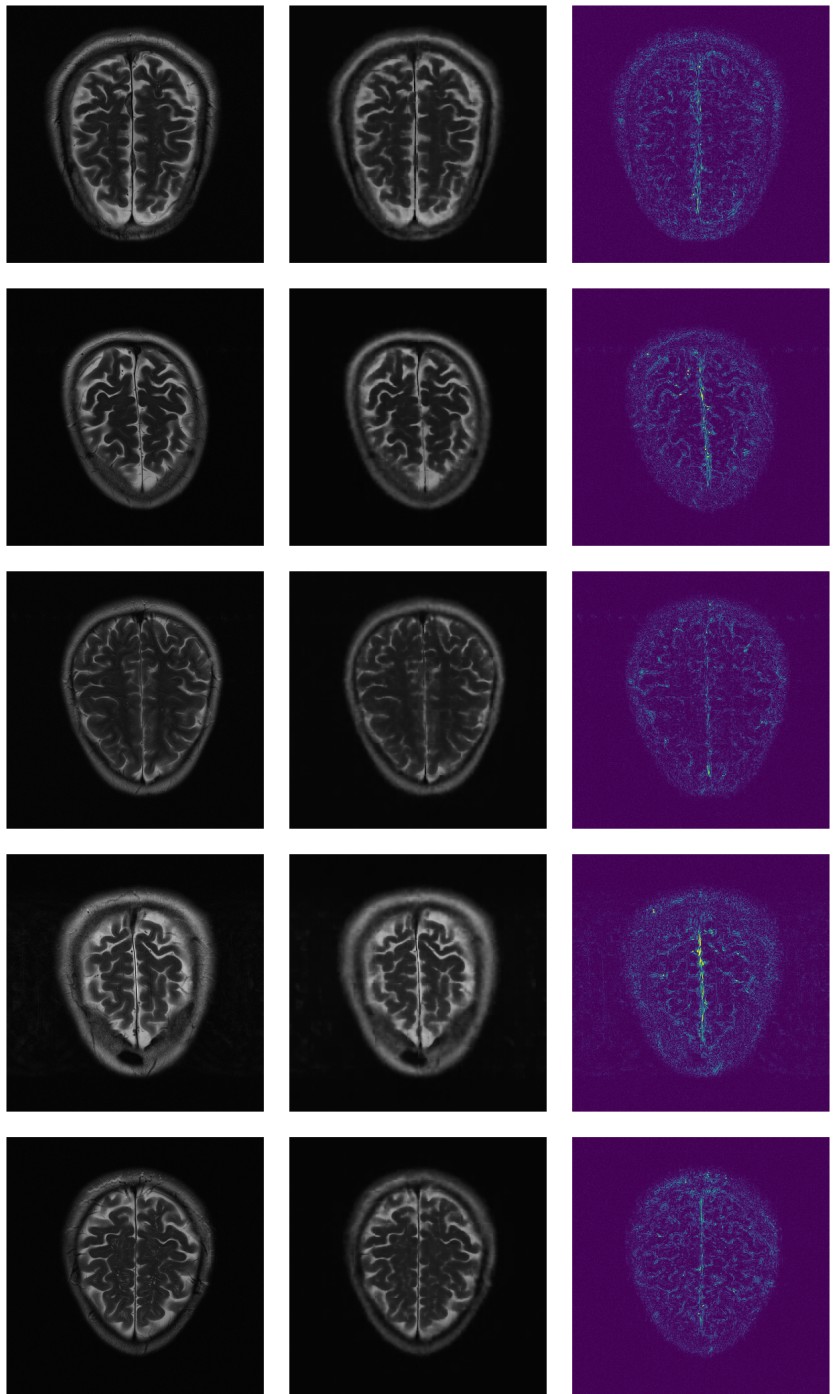

Figure 18: Qualitative compression artifacts on FastMRI using the latent dim 128 model with 0.168 bpp (original in first column, COIN++ in second column and residual in third column).

