# OpenReview forum: "COIN++: Neural Compression Across Modalities"
_TMLR — Accepted by TMLR_

### Review · Reviewer_sNx8 · 2022-10-21

**Summary Of Contributions:**

This paper introduces COIN++, an improvement of the methods introduced by COIN.
COIN++ is a implicit neural representation method, where individual data points are encoded in neural network weights (i.e., via overfitting the data).
Both COIN and COIN++ use sinusoidal layers (SIRENS).
COIN++ improves COIN’s encoding/training time by adding meta-learning, it improves COIN’s storage redundancy by factoring the COIN network into a shared “stem” and a set of modulations, and it improves COIN’s lack of quantization/entropy coding by implementing them.
COIN++ is evaluated on a range of data modalities, covering natural images, audio, MRI scans, and climate data (the latter two being 3D).
COIN++ is much faster at encoding than COIN and results in competitive (though not optimal) compression performance across experiments, demonstrating that a single neural compression method can have general and practical utility.

**Audience:**

Yes

**Broader Impact Concerns:**

None.

**Claims And Evidence:**

Yes

**Requested Changes:**

* [Critical] The information density of some of the captions seems unnecessarily low. Make the captions summarize key observations, such that the figure is more self-contained. I think, for example, mentions of the appendix are not necessary in the caption, but conclusions should be in the caption. Figure 5, in particular, has the conclusion of "improving PSNR by 2dB with the same number of parameters" in the text while the caption is mostly restating the labels on the plots. Similarly, the rate-distortion plots are captioned as rate-distortion plots without a conclusion.
* [Critical] From a presentation point of view, it should be clear what the space usage calculation is. Maybe I missed something, but the assumptions about the setting under study are not apparent e.g., how shared model weights are considered.
* [Would Strengthen] Directly measuring and ablating factors of encoding/decoding speeds (i.e., wallclock time and not iteration count) and calculations involving space usage. Figure 7d is a good start but it's not a complete characterization.
* [Would Strengthen] Adding additional metrics e.g., SSIM to evaluations. PSNR is strongly related to the (MSE) objective function used and is not a typical end-goal of compression in itself.

**Strengths And Weaknesses:**

**Strengths:**
* Motivation is clear.
* I found the paper to be easy to follow in terms of writing and content.
* The paper outlines a general method for compressing many modalities of data.
* Experiments are within scope of the claims.

**Weaknesses:**
* I found some of the captions on the figures to be lacking a conclusion or analysis.
* References to encoding speed are somewhat vague. For example, the number of iterations is used as a proxy for speed, but the time per iteration or any constant-time preprocessing (e.g., an up-front meta-learning cost) are not directly addressed. Decoding time is not addressed.
* Questions about the constants in space usage (e.g., the shared stem) are also not clear. It’s not clear what weights are considered part of the bit stream. For example, if the shared parts are not part of the bit stream due to having a single “fixed” shared model, then it’s not clear if this “fixed” model (along with the modulations) will degrade in performance once the overfitting capacity is stretched. In that case, either a new model is re-transmitted (incurring bitstream cost) or distortion increases.’
* The primary metric is PSNR but there are other metrics, such as SSIM that could have also been used.

---

> ### Author Response · Authors · 2022-10-31
> **Thank you for your review**
>
> Thank you for your thorough review! We appreciate the detailed comments and feedback which have helped us improve the paper. Below, we outline the changes we have made to address the concerns brought up in the review. Note that all the changes we have made are shown in blue in the updated paper.
>
> - *Captions*. We have updated the figure captions with more detailed information, including a summary of key observations as well as the conclusions drawn from the results in the figures. Thank you for pointing this out - making the figures more self-contained has definitely improved the clarity of the paper.
>
> - *Space usage*. We agree that the space usage calculations could be described in more detail. As such, we have added a paragraph in the paper (Section 2.3) outlining this in detail. In particular, we describe how the shared model weights are handled and exactly what is transmitted in the bit stream. We have also included a discussion of the size of the shared network and the modulations in the appendix.
>
> - *Encoding/decoding time*. We have added a table (Table 1) with both encoding and decoding time (in terms of wall clock time) for BPG, COIN and COIN++. We have also added a paragraph in the results section discussing encoding and decoding speeds, as well as the tradeoffs between various methods.
>
> - *Additional metrics*. We agree that including additional metrics such as SSIM is useful. Including this will require a bit more time as we need to rerun evaluation scripts for all experiments. We will add the results to the updated paper once we have them.
>
> Thank you again for your review - we hope the updated paper addresses your concerns, but do let us know if any additional changes are needed!

---

### Review · Reviewer_DSnf · 2022-10-22

**Summary Of Contributions:**

The authors consider using implicit neural representations (INRs) for compressing data. INRs make use of the observation that data can be represented as a function that maps a location to some value, e.g. in the case of images mapping a pixel to its RGB value. One overfits a neural network (NN) to this function, which becomes a lossy compressed representation of the original data, called the INR. INRs are a promising approach for neural data compression, but encoding a single data point is extremely expensive, as it requires fitting a neural network.

To reduce the heavy computational burden, the authors suggest meta-learning a base INR, which is way cheaper to adapt to a given data point. The adaptation is performed by learning a shift for the pre-activations of the INR. The compressed representation is thus reduced to these learned shifts, which the authors call "modulations". The authors discuss how to further reduce the codelength of the representations by quantizing and entropy coding them.

The authors test their method by compressing small and large image data, climate data on a sphere, MRI scans and audio data and demonstrate large gains compared to previous INR-based methods. Their method shows promising performance compared to state-of-the-art (SOTA) methods for each data type.

**Audience:**

Yes

**Broader Impact Concerns:**

I don't believe that a Broader Impact Statement is required.

**Claims And Evidence:**

No

**Requested Changes:**

For most necessary changes, see the "Weaknesses" section above. The authors would need to address all concerns I outline there for me to recommend acceptance.

Additionally:
 - MAML, as an abbreviation, is undefined.
 - Several citations are badly bracketed.


**Strengths And Weaknesses:**

## Strengths
I think the core idea of using meta-learning to reduce the computational burden of using INRs for compression is sound, and demonstrating that it works reasonably well without much tuning is very promising and a good contribution. Moreover, the paper contains a lot of empirical verification of the idea and has a reasonable amount of ablation study, making it quite appealing.

## Weaknesses
Unfortunately, the presentation of the ideas is poor and confusing, with some dubious claims.
 - I don't think the authors' argument that INR-based methods are more easily applicable to different data types than other neural methods is backed-up in any way. E.g. the authors claim that: "These strong results highlight the versatility of the COIN++ approach: unlike traditional codecs and autoencoder based methods (which would require spherical convolutions for the encoder), we can easily apply our method to a wide range of data modalities, including data lying on a manifold". I appreciate that INRs can be applied more easily in the sense that they don't require specialized layers (e.g. spherical convolutions), but I don't see how this is an advantage. In every experiment, the authors show that without fine-tuning the INR (e.g. its architecture) for a particular data type, their method is outperformed by any of the more specialized compression methods. This surely demonstrates that at least there is good potential in using specialized layers for INRs. In case the authors would like to retain this claim in their paper, they need to show that their method outperforms at least some sort of simple neural baselines on less conventional data types (climate data, MRI). They only compare against traditional codecs that were not designed to compress these data types. Otherwise, this claim should be removed from the paper.
 - I found Figure 1 misleading. It makes it seem like the authors are proposing to simultaneously meta-learn across different data types, which I believe is not the case. If this is a misunderstanding on my part, then the authors need to clarify the writing because there is no further implication in the paper that they would be performing the meta-learning for all data types simultaneously.
 - I couldn't parse Figure 2. The component with the latent modulation (green) is not connected to the rest of the figure. Also, there are two blue components in the figure, so I'm not sure what is mapped to what. Please redesign this figure or remove it.
 - (✓) The left plot of Figure 4 is confusing. I believe that $\theta^*$ is the only quantity that is ever used during test time, so it is not clear what the point of depicting $\theta$ is. I believe the plane shown is the parameter space, but this is not mentioned anywhere. Furthermore, connecting the $\phi$ points to $\theta^*$ is confusing. In general, I don't see how this plot is supposed to aid in understanding the ideas in the paper. Could the authors clarify this for me please?
 - The label sizes in Figures 6-10 need to be increased as they are currently hard to read.
 - (✓) What are the plots in Figure 5 depicting? The performance during training or test time?
 - (✓) In Figure 7d) the comparison appears to be tenuous, as I believe the compression is done using a GPU for the neural methods, while BPG only requires a CPU.
 - (✓) The symbol $n$ appears to be overloaded: it represents both the input size (e.g. in Eq 1 & 3) and the width of the modulated MLP in the discussion following Eq 3.
 - In the paragraph following Eq 2, the authors claim: "Secondly, as each datapoint $d$ is fitted with a separate neural network $f_\theta$, there is no information shared across datapoints. This is clearly suboptimal: natural images for example share a lot of common structure that does not need to be repeatedly stored for each individual image". I am not quite sure what the authors mean by suboptimal, but this claim is most likely false. For single-shot image compression, optimizing the INRs for individual data points is the optimal thing to do, at least in theory. Amortizing the representation across multiple images should only be optimal when considering compressing multiple images simultaneously.
 - (✓) The section describing how $\theta$ is learned using meta-learning is confusing. It appears that the authors use the learning rules in Eqs 7 and 8, but I believe they only use them in Eqs 9 and 10.
 - (✓) The "Entropy coding" paragraph in Section 4.2 is entirely dedicated to describing a plot from the paper's appendix. This figure should be moved to the main body of the paper. Otherwise, the paragraph does not have a place.

## Questions:
 - (✓) How much does patch size matter for compressing large-scale images?

---

> ### Author Response · Authors · 2022-10-31
> **Thank you for your review (1/2)**
>
> Thank you for your review! We appreciate the thorough comments and feedback. Below, we outline the changes we have made to address the concerns brought up in the review. Note that all the changes we have made are shown in magenta in the updated paper.
>
> > I don't think the authors' argument that INR-based methods are more easily applicable to different data types than other neural methods is backed-up in any way.
>
> We want to clarify that our goal is to demonstrate that INR-based methods are more easily *applicable* to different modalities than other neural methods, but not that they necessarily would outperform neural methods specialized to each data modality. We do indeed back up the claim of applicability by performing a large number of experiments on a wide and diverse set of data modalities, showing that our method is easily applicable across these data modalities, while achieving reasonable performance. Indeed, in our codebase we need to change only *two* lines of code to apply our method to a different modality (the coordinate definition and output dimension). This is in stark contrast to other neural methods, which require designing specialized layers for each data modality (which can be extremely complicated, e.g. spherical convolutions require results from group and representation theory). Further, standard neural compression methods also make use of soft quantization steps and hyperpriors, which may also lead to more difficult training when combined with specialized layers. However, it is very much possible that such a model, with enough work, would outperform COIN++ on the climate data. However, our point is exactly that it takes a lot of work to apply neural methods to different modalities, whereas this is trivial with COIN++. We have added a sentence in the climate results section to clarify this point.
>
> We also want to note that, as we try to make clear throughout the paper, our method does not achieve state of the art results in terms of rate distortion. However, as noted above, compared to other neural methods our method is much easier to apply across a wide range of modalities. The hope is then that improved performance in INR-based compression could lead to improvements in compression on a diverse set of modalities. Indeed, as highlighted in the future work section, we used a very simple setup (in terms of model structure, quantization and entropy coding) and we believe there are many directions for improving the performance of INR-based methods.
>
> > I found Figure 1 misleading. It makes it seem like the authors are proposing to simultaneously meta-learn across different data types, which I believe is not the case.
>
> It is indeed not the case that we meta-learn across different data types. We have updated the figure caption to make this clear.
>
> > I couldn't parse Figure 2. The component with the latent modulation (green) is not connected to the rest of the figure. Also, there are two blue components in the figure, so I'm not sure what is mapped to what. Please redesign this figure or remove it.
>
> We have added arrows to indicate that the output of the bottom network is passed as modulations to the main network. We have also updated the caption to describe this in more detail.
>
> > The left plot of Figure 4 is confusing.
>
> We have updated the caption to clarify the figure. As is typical in MAML papers, we show a plot of the parameter plane, where the random initialization $\theta$ is shown alongside the meta-learned initialization $\theta^*$, with a solid line indicating how the parameter moves during training. For standard MAML, dashed lines in the parameter plane itself would then indicate the rapid fitting at test time. However, in our case the fitting happens on *different* parameters $\phi$, hence why the dashed lines now reach out of the plane rather than staying in the plane. We hope the updated caption makes this clearer.
>
> > The label sizes in Figures 6-10 need to be increased as they are currently hard to read.
>
> We have increased the label sizes on Figures 7-10. Note that the CIFAR10 images in Figure 6 are too small to increase the label size without causing overlap.
>
> > What are the plots in Figure 5 depicting? The performance during training or test time?
>
> This depicts the performance on an evaluation dataset as training progresses. We have updated the caption to make this clear.

---

> > ### Author Response · Authors · 2022-10-31
> > **Thank you for your review (2/2)**
> >
> > > In Figure 7d) the comparison appears to be tenuous, as I believe the compression is done using a GPU for the neural methods, while BPG only requires a CPU.
> >
> > It is standard practice in neural compression to run neural codecs on GPU and standard codecs on CPU. Indeed, one large advantage of neural codecs is exactly that they can be accelerated on GPUs and the hope is that in the future most devices (e.g. smartphones) will have such an accelerator. However, we agree that even though it is standard practice, it is not an apples to apples comparison and we have updated the text (Section 4.2) to describe this in detail.
> >
> > > The symbol $n$ appears to be overloaded: it represents both the input size (e.g. in Eq 1 & 3) and the width of the modulated MLP in the discussion following Eq 3.
> >
> > Good catch. We have updated this to use $k$ for the MLP width.
> >
> > > In the paragraph following Eq 2, the authors claim: "Secondly, as each datapoint $\mathbf{d}$ is fitted with a separate neural network $f_\theta$, there is no information shared across datapoints. This is clearly suboptimal: natural images for example share a lot of common structure that does not need to be repeatedly stored for each individual image". I am not quite sure what the authors mean by suboptimal, but this claim is most likely false. For single-shot image compression, optimizing the INRs for individual data points is the optimal thing to do, at least in theory. Amortizing the representation across multiple images should only be optimal when considering compressing multiple images simultaneously.
> >
> > What we mean by this claim is that, when we have access to a dataset instead of a single datapoint (which we typically do, e.g. in the case of images), single shot image compression is suboptimal. Indeed, even if we have a large dataset of natural images, COIN cannot take advantage of the shared information across the images and has to "start from scratch" for every image it compresses. This is suboptimal, as there is a lot of shared structure in natural images that need not be repeatedly stored. This is why we use the expression "shared acrossed datapoints" to highlight that we are considering the case where we have access to multiple datapoints.
> >
> > > The section describing how $\theta$ is learned using meta-learning is confusing. It appears that the authors use the learning rules in Eqs 7 and 8, but I believe they only use them in Eqs 9 and 10.
> >
> > Eqs 7 and 8 describe standard MAML, while Eqs 9 and 10 describe the learning rules for the meta-learning setup we use in our paper, where the meta-learned parameters and the parameters fitted at test time are different (unlike the standard MAML case where they are the same).
> >
> > > The "Entropy coding" paragraph in Section 4.2 is entirely dedicated to describing a plot from the paper's appendix. This figure should be moved to the main body of the paper. Otherwise, the paragraph does not have a place.
> >
> > We have moved the figure from the appendix to the main body of the paper.
> >
> > > How much does patch size matter for compressing large-scale images?
> >
> > We have added a discussion of the effects of patch size in the appendix. In general, we found 32x32 to be a good tradeoff between compression rate and training/optimization time, given our GPU constraints.
> >
> > > MAML, as an abbreviation, is undefined.
> >
> > We have added a definition of the MAML abbreviation.
> >
> > > Several citations are badly bracketed.
> >
> > We carefully went over the paper again and corrected all badly bracketed citations.
> >
> > Thank you again for your review - we hope our response clarifies your questions and concerns, but please let us know if you have any more!

---

### Review · Reviewer_U5rU · 2022-10-24

**Summary Of Contributions:**

The authors propose to tackle the problem of data compression with the goal of having one method that is works across moalidites. To do so, the authors build on the recent COIN method that shifts away from the traditional encoder/decoder for data compression. Instead, the idea is to train a network on a proxy task (such as predicting pixel values from pixel indexes) and to use the network's weights as compressed representation.

More specifically, the authors adress two shortcomings of COIN, namely slow compression time (requiring network training) and lack of shared structure (training network from scratch to compress two images from the same database). Namely, the authors propose to learn a common network that is particularized to each data point using modulations (data-specific learnt vectors) that are injected at specific layers (instance-adaptation like). They reduce training time by leveraging meta-learning (MAML) and quantize and entropy-code the modulations.

The authors present a wide set of experiments (image, climate and medical data) demonstrating that COIN++ performs well among multimodal methods while stil not on a par with data type-specific compression methods (such as image compression).

**Audience:**

Yes

**Broader Impact Concerns:**

Broader impact statement seems not to be present in the main paper. No particular concerns for this work.

**Claims And Evidence:**

Yes

**Requested Changes:**

No critical changes are required for acceptance. A minor change to strengthen the paper would be as follows:
- The authors rightfully state that the modulations are easier to compress than the original models. This echoes some work in Federated Learning showing that it's generally easier to compress model deltas (that are client specific, ie the equivalent of the modulations) than models themselves. Adding some references on this could help substantiate this claim (for instance [1]).

"Reconciling Security and Communication Efficiency in Federated Learning", Prasad et al.

**Strengths And Weaknesses:**

Strengths:
- The authors clearly state the tradeoff between specific methods (such as image compression) versus multimodal compression: the proposed COIN++ method performs well but does not bridge the gap with specific methods, trading off a wide range of applicability for some loss in performance.
- The method is clearly exposed and well-executed. We thank the authors for providing experimental details in Section 4 and code in the appendix (the reviewer did not execute the code). A lot of experiments acorss modalities support the main claims of the paper.
- The paper is well written and easy to follow. In particular, the authors leverage previous work and clearly expose ths underlying methods such as MAML.

In sumamry, I would say that the paper is well written and that the method is well executed. The claims are matched with convincing experimental results. Hence, this work seems to be a good candidate for acceptance.

---

> ### Author Response · Authors · 2022-10-31
> **Thank you for your review**
>
> Thank you for your thorough and helpful review! Thank you also for highlighting the interesting connection to Federated Learning which we were not aware of. We will read the provided reference and add a discussion where appropriate. Thank you again for your review!

---

### Decision · Action_Editors · 2022-11-24

**Recommendation:** Accept with minor revision

**Comment:**

All the reviewers and myself agree that the idea is sound: applying meta learning to solving the slow compression issue of COIN, also sharing part of the architectures across data can allow better representation learning for the base network.

However there are also some suggestions in the reviews that I suggest the authors to include them in final revision:
1. Clarify the claim better in terms of "implicit neural representation based compression methods are easier to apply to multiple types of data";
2. Include a discussion of codecs on manifolds to better situate the climate experiments in the literature;

**Audience:**

People in the following fields:
Neural data compression
Meta learning
Implicit neural representations

**Claims And Evidence:**

This paper tackles the challenge of neural data compression and the approach is based on implicit neural representations.

The original COIN approach represents data as a function, e.g. for images it would use a function that maps a pixel position to its RGB value. Then for each data COIN train a neural network to overfit to the corresponding, and then use that neural network to read out the data given a grid of input positions.

This paper proposed COIN++, the idea is to "meta-learn" the parameters of the neural network using MAML, so that for each test data, one only needs to adapt the modulation part of the neural network a little bit to achieve "overfitting" to the data.

The authors present a wide set of experiments demonstrating that COIN++ performs well overall, but this method is not on a par with state-of-the-art neural lossy compression methods on images.

---

> ### Author Response · Authors · 2022-11-30
> **Final revision updates**
>
> Thank you for your feedback! Based on the requested revisions, we have made the following updates to the camera ready paper (in addition to the updates we made for the first revision):
>
> - We have updated the text to better clarify the claims of the ease of applicability of INR based compression compared to other neural compression methods (abstract, Section 1, Section 4.2, Section 4.3, Section 5)
> - We have included a discussion of codecs on manifolds to better situate the climate experiments in the literature (Section 4.3)
> - We have updated Figure 1 and its caption to improve clarity
> - We have updated Figure 2 and its caption to improve clarity
> - We have added a link to the code both in the paper and on OpenReview
>
> Thank you!